# R5: Rule Discovery with Reinforced and Recurrent Relational Reasoning

**Shengyao Lu**[1*]**, Bang Liu**[2,3*]**, Keith G. Mills**[1]**, Shangling Jui**[4]**, Di Niu**[1]
[1]Department of Electrical and Computer Engineering, University of Alberta
[2]RALI & Mila, Université de Montréal, [3]Canada CIFAR AI Chair
[4]Huawei Kirin Solution
{shengyao,kgmills,dniu}@ualberta.ca
bang.liu@umontreal.ca,jui.shangling@huawei.com

## Abstract

Systematicity, i.e., the ability to recombine known parts and rules to form new sequences while reasoning over relational data, is critical to machine intelligence. A model with strong systematicity is able to train on small-scale tasks and generalize to large-scale tasks. In this paper, we propose R5, a relational reasoning framework based on reinforcement learning that reasons over relational graph data and explicitly mines underlying compositional logical rules from observations. R5 has strong systematicity and being robust to noisy data. It consists of a policy value network equipped with Monte Carlo Tree Search to perform recurrent relational prediction and a backtrack rewriting mechanism for rule mining. By alternately applying the two components, R5 progressively learns a set of explicit rules from data and performs explainable and generalizable relation prediction. We conduct extensive evaluations on multiple datasets. Experimental results show that R5 outperforms various embedding-based and rule induction baselines on relation prediction tasks while achieving a high recall rate in discovering ground truth rules.

## 1 Introduction

While deep learning has achieved great success in various applications, it was pointed out that there is a debate over the problem of systematicity in connectionist models (Fodor & Pylyshyn, 1988; Fodor & McLaughlin, 1990; Hadley, 1994; Jansen & Watter, 2012; Dong et al., 2018). To concretely explain systematicity (Hupkes et al., 2020), let us consider the kinship problem shown in Figure 1. By training on the small-scale observed examples of family relationships where the rule paths are short (in Figure 1a&b, both of the rule paths have the length of 2), our objective is to extract the underlying rules as general principles and generalize to large-scale tasks where the rule paths are long (in Figure 1c, the rule path has the length of 3). The rules in the training examples are a) $Mother(X,Y) \leftarrow Mother(X,Z), Sister(Z,Y)$, b) $Grandma(X,Y) \leftarrow Mother(X,Z), Father(Z,Y)$. And the necessary rule in order to conclude the relation between *Mary* and *Ann* is c) $Grandma(X,Y) \leftarrow Mother(X,U), Sister(U,Z), Father(Z,Y)$, which does not explicitly show up in the training examples. Successfully giving prediction in c) shows the ability of systematicity, i.e., model's ability to recombine known parts and rules to form new sequences.

Some attempts on the graph relational data grounded in Graph Neural Networks (Schlichtkrull et al., 2018; Sinha et al., 2020; Pan et al., 2020) rely on learning embedding vectors of entities and relations, and have shown some capability of systematicity, However, since the rules are implicitly encapsulated in the neural networks, these models are lack of interpretability.

Inductive Logic Programming (ILP) (Muggleton, 1992; Muggleton et al., 1996; Getoor, 2000; Yang et al., 2017; Evans & Grefenstette, 2018) is a subfield of symbolic rule learning, which naturally learns symbolic rules and provides interpretable explanations for labels prediction as well as being able to generalize to other tasks. A traditional ILP system learns a set of rules from a collection of positive and negative examples, which entails all the positives and none of the negatives. However,

---

*Equal contribution.

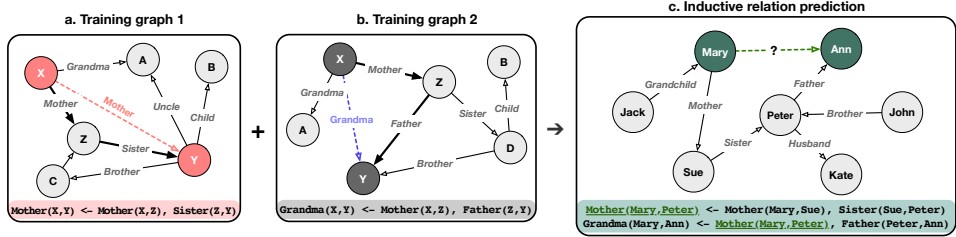

Figure 1: Illustration of rule extraction and relational reasoning over family relationship graphs.

these methods face the challenge that the search space of the compositional rules is exponentially large, making it hard to scale beyond small rule sets.

Extending the idea of ILP, Neural-Symbolic Learning (Garcez et al., 2015; Besold et al., 2017) seeks to integrate principles from neural networks learning and logical reasoning, among which Neural Logic Machines (Dong et al., 2018; Zimmer et al., 2021) and Theorem Provers (Rocktäschel & Riedel, 2017; Minervini et al., 2020a;b) aim to overcome the problems of systematicity. On one hand, the Neural Logic Machines approximate the logic predicates using a probabilistic tensor representation, and the conclusion tensors are generated by applying the neural operators over the premise tensors. These approaches have done a great job in reasoning the decision-making tasks including sorting tasks, finding the shortest paths, and so on. However, in terms of the relational reasoning tasks on graphs, Neural Logic Machines reason only the single relation prediction tasks, i.e. determining whether a single relation exists between all the queried entities or entity pairs. On the other hand, Theorem Provers jointly learn representations and rules from data via backpropagation, given a pre-defined task-specific rule template and a set of possible rules following the template. Theorem Provers are able to solve and reason the relation prediction tasks instead of only determining the existence of a single relation. However, the performance is not satisfying.

In this paper, we focus on model's systematicity for relational reasoning over graph relational prediction tasks and present a new reasoning framework named **R5**, i.e., **R**ule discovery with **R**einforced and **R**ecurrent **R**elational **R**easoning, for rule induction and reasoning on relational data with strong systematicity. R5 formulates the problem of relational reasoning from the perspective of sequential decision-making, and performs rule extraction and logical reasoning with deep reinforcement learning equipped with a dynamic rule memory. More concretely, R5 learns the short definite clauses that are in the form $u \leftarrow p_i \wedge p_j$. Since long Horn clauses (Horn, 1951) can be decomposed into short Horn clauses, a long definite clause $outcome \leftarrow p_0 \wedge p_1...p_i \wedge p_j...p_k$ used in prediction tasks is represented with a sequence of short definite clauses while performing decision-making, i.e. $p_i \wedge p_j$ is replaced by $u$. Specifically, we make the following contributions:

First, R5 performs explicit reasoning for relation prediction via composition and achieves explainability. Instead of learning embeddings for entities and relations, (Hamilton et al., 2017; Schlichtkrull et al., 2018; Wang et al., 2019; Pan et al., 2020) and performing implicit reasoning, we perform explicit relational reasoning by modeling it as sequential decision-making. Specifically, given a query (usually consists of two queried entities) and the related relationship graph, the agent recurrently selects a relation pair $r_i, r_j$ from the input graph to combine into a compound relation $r_k$ and updates the graph, until the target relation for the query is reached. Trained by reinforcement learning, the agent learns to take actions, i.e., which pair of relations to combine, given the state representations of all possible pairs of relations.

Second, we propose a dynamic rule memory module to maintain and score candidate rules during the training of the reasoning agent. Each item in the rule memory is a candidate rule in the format of $r_k \leftarrow r_i, r_j$. $r_i, r_j$ serve as the key, and $r_k$ serves as the value. In each step of decision-making, the agent queries the rule memory to re-use already stored rules for reasoning, i.e., deducing compound relations, or insert new candidate rules into it if no existing rule matches the agent's action. Each rule is associated with a score that indicates the confidence for the rule. The rules and their scores in the memory are dynamically updated during training. Finally, a set of candidate rules with scores above a threshold are kept. By recurrently applying the learned rules, R5 demonstrates strong compositional generalization ability (i.e. systematicity) in relational reasoning tasks.

Third, rather than only taking the observed relations into account, we introduce extra *invented* relations into the reasoning process. For example, R5 may learn to combine $r_1$ and $r_2$ by a rule $r \leftarrow r_1, r_2$, where $r$ is an intentionally introduced relation. The invented relations can appear on both the left or right side of a rule clause. Such a design enables our model to learn intermediate relations that do not explicitly appear in the training data to model underlying complex relations between entities, which are sometimes necessary to discover complex rules in the relational data.

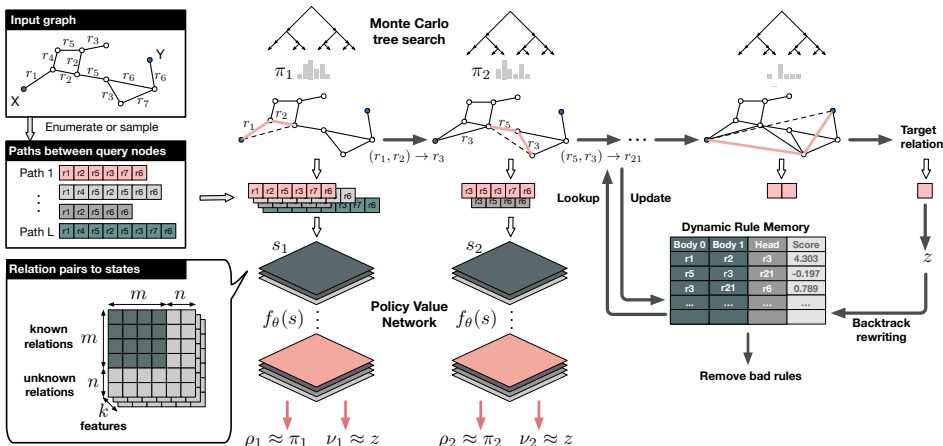

Figure 2: Overview of the propose R5 framework for rule-inductive relational reasoning.

Furthermore, we design a *backtrack rewriting* mechanism that replaces an invented relation with an observed one when R5 finds they are actually equivalent. Our goal is to explain the observed data with rules that are as simple as possible. Backtrack rewriting merges redundant relations and rules to enforce the principle of Occam's razor, which prevents overfitting and improves the robustness of our approach.

We perform extensive evaluations based on two public relation prediction datasets, CLUTRR (Sinha et al., 2019) and GraphLog (Sinha et al., 2020), and compare R5 with a variety of baseline methods. The experimental results demonstrate that our approach significantly outperforms state-of-the-art methods in terms of relation prediction accuracy and recall rate in rule discovery. Moreover, R5 exhibits a strong ability of compositional generalization and robustness to data noise. The implementation is available at `https://github.com/sluxsr/r5_graph_reasoning`.

## 2 PROPOSED APPROACH

In this section, we introduce our Rule Discovery with Reinforced and Recurrent Relational Reasoning framework, namely R5, to solve the inductive relation prediction problem. We first formally define the *relation prediction* problem discussed in this paper. Let $p_{\text{data}}(\mathcal{G}, \boldsymbol{q}, \boldsymbol{a})$ be a training data distribution, where $\mathcal{G}$ is the set of training graphs, $\boldsymbol{q} = (X, Y)$ is a query, and $\boldsymbol{a} = r$ is the answer. The graphs consist of nodes in a set $\mathcal{N}$, which are connected by relations in a set $\mathcal{R}$. $X, Y \in \mathcal{N}$ are nodes, and $r \in \mathcal{R}$ is a relation. Given $\mathcal{G}$ and the query $\boldsymbol{q}$, the goal is to predict the correct answer $\boldsymbol{a}$. Relational prediction task is actually program induction, which is a hard problem that has been studied for many years in the area of Inductive Logic Programming and Statistical Relational Learning, especially for large-scale tasks

Table 1: Notations

| | |
|---|---|
| $M$ | The set of known relations types |
| $N$ | The set of invented relations types |
| $r$ | a relation type |
| $q_j$ | a query |
| $L$ | number of paths to sample for each query |
| $L_j$ | actual number of paths for $q_j$ |
| $a_i$ | The $i^{th}$ action in in the sequence |
| $a_{B,i}$ | The $i^{th}$ relation in $a$'s body |
| $a_H$ | $a$'s head |
| $s_i$ | The $i^{th}$ current state in in the sequence |
| $z_i$ | The $i^{th}$ reward in in the sequence |
| $D_{rl}$ | Dictionary of rules |
| $D_{rls}$ | Dictionary of rules scores |
| $c$ | an entry in $D_{rls}$ |
| $B_{\text{unkn}}$ | Buffer of unused invented relation types |
| $v_i$ | a score value |

in noisy domains. Figure 2 shows an overview of R5. Our framework takes a relational graph as input, and outputs the relationship between two queried entities based on extracted rules. R5 first transforms a relation graph into a set of paths connecting the queried node (entity) pairs. After that, it recurrently applies learned rules to merge a relation pair in the paths to form a compound relation, until it outputs a final relation between the two queried nodes. The reasoning agent is trained with deep reinforcement learning and MCTS, while a dynamic rule memory module is utilized to extract rules from observations during training. The notations we used in this paper are summarized in Table 1. Note that an action $a := a_H \leftarrow (a_{B,0}, a_{B,1})$ and a rule $rl := rl_H \leftarrow (rl_{B,0}, rl_{B,1})$ share the same structure, where different $a$ and $rl$ are all relations. $a_H$ and $a_B$ represent the head and body in a rule, respectively. Next we introduce R5 in detail.

### 2.1 RECURRENT RELATIONAL REASONING

**Path sampling.** To predict the relation between two entities in graphs, we preprocess the data by sampling paths that connect the two entities from the graph. When the total number of paths is small, we enumerate all the paths. Otherwise, we randomly sample $L$ paths at maximum. Each path purely

consists of relations. For instance, in Figure 2, given the input graph, we can get 4 paths between query nodes X and Y, including $r_1-r_2-r_5-r_3-r_7-r_6$ and so on.

**Reasoning as sequential decision-making.** Our method solves the problem of relational reasoning in terms of sequential decision-making. Specifically, we train a reasoning agent based on a policy value network combined with MCTS (Silver et al., 2017a) to recurrently reason over the extracted relation paths. The policy value network $f_\theta(s)$ is a neural network with parameters $\theta$. It takes the current state $s$ as input, and outputs the action probability distribution $\boldsymbol{\rho}$ and a state value $\nu$. MCTS utilizes the policy network $f_\theta(s)$ to guide its simulations, and outputs a vector $\boldsymbol{\pi}$ representing the improved search probabilities of the available actions. The policy network $(\boldsymbol{\rho}, \nu) = f_\theta(s)$ is then trained to minimize the error between the predicted state value $\nu$ and the reward $z$ received after an episode, as well as maximize the similarity between two probability distributions $\boldsymbol{\rho}$ and $\boldsymbol{\pi}$. The loss function $l$ is $l = (z - \nu)^2 - \boldsymbol{\pi}^\mathsf{T} \log \boldsymbol{\rho} + a \|\theta\|^2$, where $a$ is a hyper-parameter.

**Action.** At each step of an episode, the MCTS outputs an action $(a_{B,0}, a_{B,1})$, which is a relation pair denoted as $a_B$. Furthermore, by looking up the dynamic rule memory, it obtains $a : a_H \leftarrow (a_{B,0}, a_{B,1})$ that contains a head relation $a_H$ to be deducted to, which means that the relation pair $(a_{B,0}, a_{B,1})$ in the path will be substitute with $a_H$. For example, in Figure 2 at step 1, MCTS outputs an action $a : r_3 \leftarrow (r_1, r_2)$, and the path $r_1-r_2-r_5-r_3-r_7-r_6$ is transformed into $r_3-r_5-r_3-r_7-r_6$. By recurrently applying different actions to a path between the query nodes, it will be transformed into a single relation at the end of an episode.

**State.** Instead of encoding the walking paths between the query nodes as other RL-based rule induction methods (Shen et al., 2018; Das et al., 2017; Xiong et al., 2017), we make use of the features of the possible relations pairs at the current state. As shown in Figure 2, we define the state $s \in \mathbb{R}^{(m+n)\times(m+n)\times k}$ as the representation of all the possible relations pairs among the current paths, where $m$ is the number of observed relations, $n$ is the number of invented relations we assumed, and $k$ is the dimension of features of each relation pair. As previously discussed, a relation pair may be deducted to an invented relation $r_* \in N$. Besides, $r_*$ can be paired with another relation $r$, and maybe further deducted to another invented relation $r'_* \in N$. Our state space design enables us to represent such complicated rules with the invented relations that serve as intermediate predicates, which is essential to model complex relational data. $n$ invented relations are allowed in our design, where $n$ depends on the complexity of input. An example is presented in Appendix A.4.

Even if sometimes the input data can be explained by rules without invented relation, the invented relations in our state design can actually help to speed up model training. For example, when we observe that a relation pair $r_1$ and $r_2$ shows up frequently in the training data, we can sample an invented relation $r_*$ as the head, and form a candidate rule $r_* \leftarrow (r_1, r_2)$. In this way, our model learns to merge $r_1$ and $r_2$ quickly without the need to figure out what exactly $r_*$ is. The value of $r_*$ will be inferred by our model with Backtrack Rewriting mechanism, which we will introduce in more detail later. Without these invented relations, a reasoning agent has to infer what exactly $r_*$ is at an early state, otherwise it cannot proceed to merge the relation pairs in a path correctly. Thus, the invented relations actually serve as a buffering mechanism to allow an agent to learn "what to merge" first, rather than inferring the exact rule at an early stage.

The features we utilized to represent each relation pair can be summarized into two groups. The first group includes the general and statistical features of the pair, such as the number of occurrences, the index of the most viewed position among the paths, the number of occurrences at the most viewed position, and the types of the two relations consist the pair. The other group includes the rules-related features, such as whether the relation pair is in the obtained rules, whether the relation pair consists of two known relations, whether the rule head is a known relation if the pair is in the rules memory, and the score of the corresponding rule if the pair is in the rules memory.

**Reward.** During an episode, actions are applied to a relation path recurrently, until the path is deducted to a single relation $r$, and a reward $z_T \in \{-1, 0, +1\}$ will be assigned to all the states in this episode. If $r$ is an known relation, but is not the target relation, $z_T = -1$. If $r$ is an invented relation, $z_T = 0$. If $r$ is the target relation, $z_T = 1$.

## 2.2    RULE INDUCTION WITH A DYNAMIC RULE MEMORY

The above recurrent reasoning agent enables us to predict the relation between two queried nodes more efficiently than only using MCTS. To extract rules from observed data, we further design

a dynamic rule memory module which stores and updates candidate rules and interact with the reasoning agent during training. Our memory design can be interpreted as two hash tables with the same keys: one hash table $D_{rl}$ in the form of $(r_{B0}, r_{B1}) : r_h$ is used to memorize the candidate rules, and another hash table $D_{rls}$ in the form of $(r_{B0}, r_{B1}) : \texttt{score}$ to track the rule scores. Note that in Figure 2, we union the two tables as they share the same keys. A newly added rule will have an initial score of 0. In Figure 2, a rule $(r_3) \leftarrow (r_1, r_2)$ was stored in the hash tables, where the rule body $(r_1, r_2)$ is the key, and the rule head $r_3$ and score 4.303 are the values in $D_{rl}$ and $D_{rls}$, respectively.

We utilize a buffer $B_{\text{unkn}}$ to store the $n$ intermediate invented relations as described in Sec. 2.1. When an invented relation type $r \in N$ is used to complete a rule, it will be removed from $B_{\text{unkn}}$. If $r$ is freed from the rule memory $D_{rl}$, then it will be added to $B_{\text{unkn}}$ again.

Recall that at each time step the MCTS will return an action body $a_B$. In order to complete the action, we get the action head $a_H$ by

$$a_H = \begin{cases} D_{rl}[a_B], & \text{if } a_B \in D_{rl}, \\ \text{RandomSample}(B_{\text{unkn}}), & \text{otherwise.} \end{cases} \quad (1)$$

Then, if $a_B \notin D_{rl}$, we update the memory $D_{rl}$ with $D_{rl}[a_B] = a_H$, and compute the score $c_a$ by

$$c_a = \begin{cases} v_0, & a_B \in D_{rl}, \forall r \in a, r \in M, \\ v_1, & a_B \in D_{rl}, a_H \in M, \exists r \in a_B, r \in N, \\ v_2, & a_B \in D_{rl}, a_H \in N, \\ v_3, & a_B \notin D_{rl}, \forall r \in a_B, r \in M, \\ v_4, & a_B \notin D_{rl}, \exists r \in a_B, r \in N, \end{cases} \quad (2)$$

where $v_0 > v_1 > 0 > v_2 > v_3 > v_4$. We can see that an action consisting of three known relations will receive the largest positive score, and an action with an invented relation in the head and at least an invented relation in the body will receive the smallest negative score. This way we penalize the usage of invented relations to prevent overfitting.

After extracted candidate rules, at the end of each training episode, we further utilize Backtrack Rewriting to consolidate all the candidate rules and replace invented relations in a rule with observed relations if we found they are equivalent. Algorithm 1 describes our Backtrack Rewriting mechanism for updating the rule memory $D_{rl}$. Specifically, assume the action sequence we have taken in an episode is $\{a_0, a_1, ...a_t, ...a_T\}$, and the head relation of the last action $a_T$ is $r := a_{TH} = D_{rl}[a_{TB}]$. As $y$ is the target relation, we can backtrack $r$ in $D_{rl}$, and replace all its occurrences with $y$. Before updating $D_{rl}[a_{TB}]$, we update the corresponding entries in $D_{rls}$ of the actions $\{a_0, a_1, ...a_t, ...a_T\}$ in the episode by

$$D_{rls}[a_{tB}] = \begin{cases} c_{a_t} + v_{T\text{pos}}, & r = y, \\ v_{T\text{neg}}, & r \in M, r \neq y, \\ c_{a_t}, & \text{otherwise,} \end{cases} \quad (3)$$

where $r := D_{rl}[a_{TB}]$, and $v_{T\text{neg}} < 0 < v_{T\text{pos}}$ are the episode end scores. Next, we decay the scores $D_{rls}[a_B]$ of all the rules $a_H \leftarrow a_B$ over episodes by:

**Algorithm 1** Backtrack Rewriting

1: series Input: $D_{rl}$, $B_{\text{unkn}}$, last action $a_T$, target relation $y$
2: $r = D_{rl}[a_{TB}]$
3: **if** $r \in N$ **then**
4:     **for** $a_B$ in $D_{rl}$ **do**
5:         **if** $D_{rl}[a_B] = r$ **then**
6:             $D_{rl}[a_B] = y$
7:     **for** $a_B$ in $D_{rl}$ **do**
8:         $a_H = D_{rl}[a_B]$
9:         **if** $a_{B,0} = r$ **and** $a_{B,1} = r$ **then**
10:             $key = (y, y)$
11:         **else if** $a_{B,0} = r$ **then**
12:             $key = (y, a_{B,1})$
13:         **else if** $a_{B,1} = r$ **then**
14:             $key = (a_{B,0}, y)$
15:         **else**
16:             *continue*
17:         **if** $key \notin D_{rl}$ **or** $D_{rl}[key] \in N$ **then**
18:             delete $a_B$ from $D_{rl}$
19:             $D_{rl}[key] = a_H$
20:         **else**
21:             delete $a_B$ from $D_{rl}$
22:     remove $r$ from $B_{\text{unkn}}$

$$D_{rls}[a_B] = \begin{cases} D_{rls}[a_B](1+\epsilon), & D_{rls}[a_B] < 0, \\ D_{rls}[a_B](1-\epsilon), & D_{rls}[a_B] > 0, \end{cases} \quad (4)$$

where $\epsilon$ is a small positive decay factor. Thus, the score of a rule that is not accessed over training episodes will decrease.

Table 2: Results on CLUTRR, trained on stories of lengths $\{2,3\}$ and evaluated on stories of lengths $\{4,...,10\}$. [*] means the numbers are taken from CTP's paper. [$_s$] means fine-tuned on short stories.

| | 4Hops | 5Hops | 6Hops | 7Hops | 8Hops | 9Hops | 10Hops |
|---|---|---|---|---|---|---|---|
| R5 | .98±.02 | **.99±.02** | .98±.03 | **.96±.05** | **.97±.01** | **.98±.03** | **.97±.03** |
| $CTP^*_L$ | .98±.02 | .98±.03 | .97±.05 | .96±.04 | .94±.05 | .89±.07 | .89±.07 |
| $CTP^*_A$ | **.99±.02** | .99±.01 | **.99±.02** | .96±.04 | .94±.05 | .89±.08 | .90±.07 |
| $CTP^*_M$ | .97±.03 | .97±.03 | .96±.06 | .95±.06 | .93±.05 | .90±.06 | .89±.06 |
| $GNTP^*$ | .49±.18 | .45±.21 | .38±.23 | .37±.21 | .32±.20 | .31±.19 | .31±.22 |
| $GAT^*_s$ | .91±.02 | .76±.06 | .54±.03 | .56±.04 | .54±.03 | .55±.05 | .45±.06 |
| $GCN^*_s$ | .84±.03 | .68±.02 | .53±.03 | .47±.04 | .42±.03 | .45±.03 | .39±.02 |
| $RNN^*_s$ | .86±.06 | .76±.08 | .67±.08 | .66±.08 | .56±.10 | .55±.10 | .48±.07 |
| $LSTM^*_s$ | .98±.04 | .95±.03 | .88±.05 | .87±.04 | .81±.07 | .75±.10 | .75±.09 |
| $GRU^*_s$ | .89±.05 | .83±.06 | .74±.12 | .72±.09 | .67±.12 | .62±.10 | .60±.12 |
| $CNNH^*_s$ | .90±.04 | .81±.05 | .69±.10 | .64±.08 | .56±.13 | .52±.12 | .50±.12 |
| $CNN^*_s$ | .95±.02 | .90±.03 | .89±.04 | .80±.05 | .76±.08 | .69±.07 | .70±.08 |
| $MHA^*_s$ | .81±.04 | .76±.04 | .74±.05 | .70±.04 | .69±.03 | .64±.05 | .67±.02 |

Table 3: Results on CLUTRR, trained on stories of lengths $\{2,3,4\}$ and evaluated on stories of lengths $\{5,...,10\}$. [*] means the numbers are taken from CTP's paper. [$_s$] means fine-tuned on short stories.

| | 5 Hops | 6 Hops | 7 Hops | 8 Hops | 9 Hops | 10 Hops |
|---|---|---|---|---|---|---|
| R5 | .99±.02 | .99±0.4 | **.99±.03** | 1.0±.02 | **.99±.02** | **.98±.03** |
| $CTP^*_L$ | .99±.02 | .98±.04 | .97±.04 | .98±.03 | .97±.04 | .95±.04 |
| $CTP^*_A$ | .99±.04 | .99±.03 | .97±.03 | .95±.06 | .93±.07 | .91±.05 |
| $CTP^*_M$ | .98±.04 | .97±.06 | .95±.06 | .94±.08 | .93±.08 | .90±.09 |
| $GNTP^*$ | .68±.28 | .63±.34 | .62±.31 | .59±.32 | .57±.34 | .52±.32 |
| $GAT^*_s$ | .99±.00 | .85±.04 | .80±.03 | .71±.03 | .70±.03 | .68±.02 |
| $GCN^*_s$ | .94±.03 | .79±.02 | .61±.03 | .53±.04 | .53±.04 | .41±.04 |
| $RNN^*_s$ | .93±.06 | .87±.07 | .79±.11 | .73±.12 | .65±.16 | .64±.16 |
| $LSTM^*_s$ | .98±.03 | .95±.04 | .89±.10 | .84±.07 | .77±.11 | .78±.11 |
| $GRU^*_s$ | .95±.04 | .94±.03 | .87±.08 | .81±.13 | .74±.15 | .75±.15 |
| $CNNH^*_s$ | .99±.01 | .97±.02 | .94±.03 | .88±.04 | .86±.05 | .84±.06 |
| $CNN^*_s$ | **1.0±.00** | **1.0±.01** | .98±.01 | .95±.03 | .93±.03 | .92±.04 |
| $MHA^*_s$ | .88±.03 | .83±.05 | .76±.04 | .72±.04 | .74±.05 | .70±.03 |

Finally, when a rule score is less than a negative threshold $\sigma$, the corresponding rule will be omitted from the rule memory. We recursively check the rule memory to remove all the candidate rules that contain the omitted rule's head $a_{Hbad} \in N$. Thus, the bad rules will be gradually removed from the rule memory. $B_{\text{unkn}}$ and $D_{rls}$ will be updated accordingly. Besides, when the invented relations buffer $B_{\text{unkn}}$ is empty, we will look up the rules scores hash table $D_{rls}$ to find the rule $a_{Hbad} \leftarrow a_{Bbad}$ that has the smallest rule score, where $a_{Hbad} \in N$. Then all the corresponding rules in $D_{rl}$ that contain $a_{Hbad}$ will be removed, and $a_{Hbad}$ will be added back to $B_{\text{unkn}}$.

## 3 EXPERIMENTS

We evaluate R5 on two datasets: CLUTRR (Sinha et al., 2019) and GraphLog (Sinha et al., 2020), which test the logical generalization capabilities. We compare with embedding-based and rule induction baselines for both CLUTRR and GraphLog.

### 3.1 SYSTEMATICITY ON CLEAN DATA

We first evaluate R5 on the CLUTRR dataset and demonstrate its Systematicity. In other words, we test whether R5 can perform reasoning over graphs with more hops than the training data.

**Dataset and Experimental Setup.** CLUTRR (Sinha et al., 2019) is a dataset for inductive reasoning over family relations. Paragraphs describing family relationships are encoded as a large set of graphs, where family members are modeled as nodes, and the relations in between are modeled as edges. The goal is to infer the relation between two family members, which is not explicitly mentioned in the paragraph. The train set contains graphs with up to 3 or 4 edges, and the test set contains graphs with up to 10 edges. The train and test splits do not share the same set of entities.

As baselines, we evaluate several embedding-based models (Veličković et al., 2017; Kipf & Welling, 2017; Schuster & Paliwal, 1997; Hochreiter & Schmidhuber, 1997; Chung et al., 2014; Kim, 2014; Kim et al., 2015; Vaswani et al., 2017) and several neuro-symbolic models that consists the rule induction baselines (Minervini et al., 2020a;b). The results of the baselines are taken from Minervini et al. (2020b). Due to limited space, for the baselines other than CTPs and GNTPs, we only present the results fine-tuned on short stories. The comparisons with these baselines fine-tuned on long stories are provided in the Appendix C, and none of them can beat R5.

**Results.** Table 10, 11 present the prediction accuracy on two datasets pulished by Sinha et al. (2019). We call the two datasets $CLUTRR_{k \in \{2,3\}}$ and $CLUTRR_{k \in \{2,3,4\}}$, as they train on graphs with $\{2,3\}$ edges while test on graphs with $\{4,...,10\}$ edges, and train on graphs with $\{2,3,4\}$ edges while test on graphs with $\{5,...,10\}$ edges, respectively. As shown in both tables, R5 either outperforms all the baselines or reach the best results in both metrics. In particular, R5 has better generalization capability than CTPs. For both datasets, the prediction accuracy of CTPs goes down when evaluated on longer stories. In contrast, R5 still predicts successfully on longer stories without significant performance degradation. This demonstrates the strong compositional generalization ability of R5, in terms of systematicity, productivity, substitutivity, and localism (Hupkes et al., 2020).

### 3.2 SYSTEMATICITY ON NOISY DATA

In Section 3.1, we test model performance with no bad cases in data. A bad case is a logically wrong example. For example, *the mother of a sister is a father* is a bad case. Apart from the compositional

Table 4: Results of reasoning on the selected GraphLog datasets (worlds). ND: number of distinct relations sequences that traverse between query nodes; ARL: Average resolution length. [*] means the numbers are taken from the original papers. [†] means we rerun the methods using the released source code.

| World ID | ND | ARL | E-GAT* Accuracy | R-GCN* Accuracy | CTP† Accuracy | CTP† Rules Recall | R5 Accuracy | R5 Rules Recall |
|---|---|---|---|---|---|---|---|---|
| World 2 | 157 | 3.21 | 0.412 | 0.347 | 0.685±0.03 | 0.80±0.05 | **0.755±0.02** | 1.0±0.00 |
| World 3 | 189 | 3.63 | 0.473 | 0.451 | 0.624±0.02 | 0.85±0.00 | **0.791±0.03** | 1.0±0.00 |
| World 13 | 149 | 3.58 | 0.453 | 0.347 | 0.696±0.01 | 0.90±0.00 | **0.895±0.00** | 1.0±0.00 |
| World 17 | 147 | 3.16 | 0.347 | 0.181 | 0.789±0.04 | 0.85±0.00 | **0.947±0.02** | 1.0±0.00 |
| World 30 | 177 | 3.51 | 0.426 | 0.357 | 0.658±0.01 | 0.80±0.00 | **0.853±0.01** | 0.95±0.00 |
| World 6 | 249 | 5.06 | 0.536 | 0.498 | 0.533±0.03 | 0.85±0.00 | **0.687±0.05** | 0.9±0.00 |
| World 7 | 288 | 4.47 | 0.613 | 0.537 | 0.513±0.03 | 0.75±0.05 | **0.749±0.04** | 0.95±0.00 |
| World 8 | 404 | 5.43 | 0.643 | 0.569 | 0.545±0.02 | 0.70±0.00 | **0.671±0.03** | 0.95±0.00 |
| World 11 | 194 | 4.29 | 0.552 | 0.456 | 0.553±0.01 | 0.85±0.00 | **0.803±0.01** | 1.0±0.00 |
| World 32 | 287 | 4.66 | 0.700 | 0.621 | 0.581±0.04 | 0.95±0.00 | **0.841±0.03** | 1.0±0.00 |

generalization ability, it is also important to assess how effectively a model can discover all underlying rules. The existing relational reasoning works only show that they are able to improve relation prediction with some learned rules, without an evaluation on whether the extracted rules are exhaustive. GraphLog, the dataset we use in this subsection, provides the ground rules used to create each logical world, so we evaluate this ability by the recall rate of ground truth rules.

**Dataset and Experimental Setup.** Graphlog (Sinha et al., 2020) is a benchmark suite designed to evaluate systemeticity. It contains logical worlds where each world contains graphs that are created using a different set of ground rules. Similar to CLUTRR, the goal is to infer the relation between a queried node pair. However, GraphLog is much more challenging than CLUTRR. Firstly, it contains more bad examples than CLUTRR does. Secondly, the graphs in both train sets and test sets of Graphlog contain more edges; it is normal that there are more than 10 edges in a graph. Thirdly, the underlying compositional rules may not be shared between a train set and its corresponding test set. The above facts make GraphLog a more challenging dataset for rule induction.

In the evaluations on GraphLog, we consider E-GAT (Sinha et al., 2020) and R-GCN (Schlichtkrull et al., 2018) as the embedding-based baselines, and CTPs (Minervini et al., 2020b) as the rule induction baseline. For E-GAT and R-GCN, we use the results in (Sinha et al., 2020). For CTPs, we use the implementation publicly provided by the author with the best configurations. In addition, we use the core generator provided by (Sinha et al., 2020), which is used to produce the GraphLog datasets, to generate three new worlds: Re 0, Re 1, Re 2. The new worlds have less training data and more distinct relations sequences. We aim to test how the rule induction models perform when only a small amount of training samples are given. We divide the 57 worlds in Graphlog into two groups in terms of the average resolution length (ARL), where the resolution length is the number of nodes traversed in the correct path between the queried nodes. Figure 1a,b are instances of resolution lengths of 2, and Figure 1c is an instance of resolution length of 3. We then randomly pick 5 worlds with an ARL less than 4, and 5 other worlds with an ARL greater than 4 to conduct experiments on.

**Results.** Table 4 shows the results on 10 selected worlds (datasets) published by (Sinha et al., 2020). When the ARL is larger, the dataset becomes noisier and contains more bad cases. In terms of relation prediction accuracy, R5 significantly outperforms both the embedding-based and rule induction baselines over all 10 datasets. In terms of the ground rule recall rate, R5 also consistently outperforms CTP, and even reaches full recall rate on some datasets.

At a closer inspection, the two GNN models perform poorly for ARL $< 4$. A dataset with a smaller ARL indicates the graphs are very simple, and models that represent the graph structure based on message-passing have not enough information to learn from. As for datasets with ARL $> 4$, the graph structures become more complex, the GNNs are able to aggregate more information to each node from the neighbors. Thus, the performance of GNNs improves on these datasets. However, CTP performs worse than GNNs in this case, due to two reasons. On one hand, the increasing amount of bad cases influences the rule recall rate, making CTP less confident on some correct rules. On the other hand, it is hard for CTP to generalize to complex data.

Table 5 presents the results for the three new datasets created that have fewer training samples. In particular, resolution lengths of graphs in Re 0 can be up to 10, while resolution lengths of graphs in Re 1 can be up to 15. Re 2 requires the models to train on graphs with resolution lengths of $\{3, ..., 5\}$, and to test on graphs with resolution lengths of $\{2, ..., 7\}$. As shown in the table, R5 achieves better results than CTP when the amount of training facts is very small. And we can observe from Re 0 and Re 1 that as the resolution length becomes longer, CTP has a sharp performance drop on the prediction accuracy, whereas R5 maintains relatively high accuracy and full ground rule recall rate.

Table 5: Results on new datasets generated using GraphLog generator. ND: number of distinct relations sequences that traverse between query nodes; ARL: Average resolution length. RL: resolution length. ACC: Accuracy. We rerun CTPs using the released source code.

| Data set | ND | ARL | Train RL | Test RL | CTP | | R5 | |
|---|---|---|---|---|---|---|---|---|
| | | | | | ACC | Recall | ACC | Recall |
| Re 0 | 631 | 3.90 | 2∼10 | 2∼10 | 0.425±0.03 | 0.75±0.00 | **0.665±0.06** | 1.0±0.00 |
| Re 1 | 736 | 5.19 | 2∼15 | 2∼15 | 0.190±0.06 | 0.75±0.00 | **0.575±0.02** | 1.0±0.00 |
| Re 2 | 725 | 3.81 | 3∼5 | 2∼7 | 0.359±0.02 | 0.30±0.05 | **0.446±0.04** | 0.70±0.00 |

In the train set of Re 2, there is no *simple* data. Simple data refers to a sample whose resolution length is 2, and the ground rules can be naturally collected from such data. However, in Re 2, as we train on samples of lengths $\{3, ..., 5\}$ and evaluate on samples of lengths $\{2, ..., 7\}$, the models have to infer the ground rules. As is shown, R5 can still extract most of the ground rules while CTP fails.

### 3.3 ABLATION STUDIES

**Policy Value Network.** As stated earlier, the policy value network is trained to sequentially decide which edge pair we should select to do a deduction. Without training this network, only MCTS will help us to make the decisions, whose action distribution is given by the numbers of visits at all possible nodes. From Table 6 we acquire that although we are still able to recall the ground rules with rules extraction, the prediction accuracy drastically decreases since the paths are not deducted in the correct order.

**Pre-allocated Invented Relations.** We pre-allocate $n$ invented relations to allow a relation pair to deduct to an intermediate predicate when it still is unclear whether the relation pair can give us a correct rule. And the intermediate predicate will be tracked during the training until it is identified as a known relation. Refer to Table 6, when we only allow a single invented relation, the prediction accuracy will drop, which indicates that the extra hidden relations improve the model's capability of learning complicated graphs. And in Re 2, where the training samples' resolution lengths are greater than 2, the results catastrophically decrease. We evaluate the prediction accuracy and rules recall each training epoch. As shown in Figure 3, the model loses the induction ability at the early stage with only one intermediate predicate allowed. Once the first ground rule is extracted, more will be gradually learned, and finally the R5 with a single hidden predicate reaches the same rules recall rate as R5 with 50 hidden predicates. This figure proves that a number of pre-allocated invented relations can not only help to learn complicated graphs, but also speed up the convergence.

Table 6: Ablation study of R5. PVN: policy value network.

| | World 2 | | World 6 | | Re 2 | |
|---|---|---|---|---|---|---|
| | ACC | Recall | ACC | Recall | ACC | Recall |
| R5 | **0.755** | 1.0 | **0.687** | 0.9 | **0.446** | 0.70 |
| R5 w/o PVN | 0.502 | 1.0 | 0.407 | 0.9 | 0.208 | 0.60 |
| R5, n=1 | 0.661 | 1.0 | 0.595 | 0.85 | 0.349 | 0.70 |

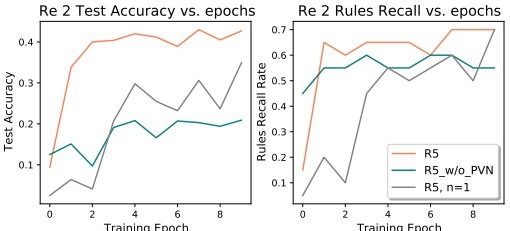

Figure 3: Ablation study of R5 on Re 2 in terms of test accuracy and rules recall rate.

## 4 RELATED WORK

**Graph Neural Networks and General Relation Prediction Tasks.** Graph Convolutional Networks (GCNs) (Bruna et al., 2014; Li et al., 2016; Defferrard et al., 2016; Kipf & Welling, 2017) is a family of message-passing neural architectures designed for processing graph data. While most GCNs operate only with the entities and binary links, some recent works (Schlichtkrull et al., 2018; Sinha et al., 2020; Pan et al., 2020) are applicable on relational data where the relation types (i.e., edge types) between entities (i.e., vertices) matters. The GCN-based models can capture lifted rules defined on the sets of entities and have presented good results on the node classification tasks (Sinha et al., 2020). However, since the lifted rules are implicitly encapsulated in the neural networks, these models are lack of interpretability.

**ILP, Neural-Symbolic Learning and Systematicity.** Inductive Logic Programming (ILP) (Muggleton, 1992; Muggleton et al., 1996; Getoor, 2000; Evans & Grefenstette, 2018) is a subfield of symbolic rule learning, which learns logic rules derived from a limited and predefined set of rule templates. A traditional ILP system learns a set of rules from a collection of examples, which entails all the positives and none of the negatives. Such a system naturally provides interpretable explanations for labels prediction as well as being able to generalize to new tasks but is hard to scale up to

the exponentially large space of compositional rules (Evans & Grefenstette, 2018; Yang et al., 2017) as the number of relations and entities increases. On the other hand, the relational path assumption suggests that there usually exist fixed-length paths of relations linking the set of terms (entities) that satisfy a target concept (Mooney, 1992). Under this assumption, Cropper & Muggleton (2015) reduces the complexity of individual rules with the latent predicates. Statistical unfolded logic (SUL) (Dai & Zhou, 2016) searches for a relational feature set from relational space and grounded relational paths, such that statistical learning approaches are applied to extract weighted logic rules. Using a statistical discriminative model, SUL also enables predicate invention.

Neural-Symbolic models (Garcez et al., 2015; Besold et al., 2017) seek to integrate principles from neural networks learning and logical reasoning, among which Neural Logic Machines (NLMs) (Dong et al., 2018; Zimmer et al., 2021) and Theorem Provers (Rocktäschel & Riedel, 2017; Minervini et al., 2020a;b) attempt to overcome the problems of systematicity. The NLMs successfully show the systematicity for up to 50 entities on some selected decision-making tasks. However, in terms of relational reasoning on graphs, NLMs only consider the *binary classification tasks*. Although the links in the graphs have various types, for each task, NLMs predict the existence of a single relation type instead of concluding which type the relation is when there is one. When performing general relation prediction, NLMs have to create a number of tasks where each task predicts for a single relation type, making the total number of tasks grows as the number of relation types increases. As a result, NLMs are hard to scale up on relation prediction tasks.

Theorem Provers, in particular the Conditional Theorem Provers (CTPs) (Minervini et al., 2020a;b), are capable of the relation prediction problem without the need to create a number of tasks. CTPs use templates equipped with a variety of selection modules to reformulate a given goal into subgoals, and jointly learn representations and rules from data. However, the depth of the templates (refer to *hops* in their code) is limited, which influences the number of parameters of the model. In contrast, as explained in Section 2.1, R5 is able to handle the complicated rules where the allowed rule depth is much larger. That is because the rule depth is only related to the number of steps in an episode of the sequential decision-making process of R5, which does not touch the number of model parameters. We have also presented in Section 3.1 and 3.2 that R5 gives better relation prediction results on the systematicity tasks and shows better performance in terms of the rules recall.

**Relational Reasoning for Knowledge Base Completion.** Many recent works propose a variety of relational reasoning models for efficient Knowledge Base Completion (KBC), which differs from what we are focusing on. These works include the embedding-based methods (Sun et al., 2019; Trouillon et al., 2017; Dettmers et al., 2017; Bordes et al., 2013), GNN-like methods (Teru et al., 2020; Mai et al., 2020), Neural-Symbolic Methods (Rocktäschel & Riedel, 2017; Minervini et al., 2020a;b; Qu et al., 2021), and RL Path-finding Agents (Xiong et al., 2017; Chen et al., 2018; Das et al., 2017; Lin et al., 2018; Shen et al., 2018). Most of them are designed for the transductive aspects instead of systematicity. Moreover, the KBC tasks can be treated as entity sorting problems, where the objective is ranking the tail entities given a query $(h, r, ?)$ with head entity $h$ and the relation $r$ are given (Wang et al., 2021). KBC is different from the general relation prediction tasks investigated in this paper. It is also non-trivial to extend our research to tackle KBC tasks. Since the relation $r$ is given in the query, rather than deciding the relation type between a head entity $h$ and a candidate tail $t*$ like what R5 might do, a better choice would be directly determining whether the single relation $r$ exists between $h$ and $t*$ like what NLMs might be able to do. Therefore, we do not discuss R5's performance on KBC tasks in this paper. Although non-trivial, surely it is worth trying to extend the main structure of R5 to KB. We leave this task to future works.

## 5 CONCLUSION

We propose R5, a framework that models relational reasoning as sequential decision-making. Our framework performs explicit and explainable relational reasoning with a design of state and action spaces based on pairs of relations. Besides, R5 performs rule induction with a dynamic rule memory module, which is updated during the training of an agent. We further incorporate hidden (invented) relations in our state and action spaces to help accelerate model training as well as improve rule mining. R5 exhibits high accuracy for relation prediction and a high recall rate for rule discovery, as has been demonstrated by extensive experimental results. We also show that R5 has a strong ability of systematicity and is robust to data noise.

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

# SUPPLEMENTARY MATERIALS

## A DATASET STATISTICS AND IMPLEMENTATION DETAILS

We have introduced R5 in Section 3. Due to the limited space, some implementation details are not covered. In this section, we explain the parameterization and implementation details of R5.

### A.1 DATASET STATISTICS

The statistics of CLUTRR are summarized in Table 7, and the statistics of GraphLog are summarized in Table 8.

Table 7: Statistics of CLUTRR datasets.

| Dataset | #relations | #Train | #Test |
|---|---|---|---|
| $CLUTRR_{k \in \{2,3\}}$ | 22 | 10,094 | 900 |
| $CLUTRR_{k \in \{2,3,4\}}$ | 22 | 15,083 | 823 |

Table 8: Statistics of GraphLog datasets. NC: number of classes. AN: average number of nodes. ND: number of distinct resolution edge sequences (distinct descriptors). ARL: average resolution length. AE: average number of edges.

| World ID | NC | ND | ARL | AN | AE | #Train | #Test |
|---|---|---|---|---|---|---|---|
| World 2 | 17 | 157 | 3.21 | 9.8 | 11.2 | 5000 | 1000 |
| World 3 | 16 | 189 | 3.63 | 11.1 | 13.3 | 5000 | 1000 |
| World 13 | 16 | 149 | 3.58 | 11.2 | 13.5 | 5000 | 1000 |
| World 17 | 17 | 147 | 3.16 | 8.16 | 8.89 | 5000 | 1000 |
| World 30 | 17 | 177 | 3.51 | 10.3 | 11.8 | 5000 | 1000 |
| World 6 | 16 | 249 | 5.06 | 16.3 | 20.2 | 5000 | 1000 |
| World 7 | 17 | 288 | 4.47 | 13.2 | 16.3 | 5000 | 1000 |
| World 8 | 15 | 404 | 5.43 | 16.0 | 19.1 | 5000 | 1000 |
| World 11 | 17 | 194 | 4.29 | 11.5 | 13.0 | 5000 | 1000 |
| World 32 | 16 | 287 | 4.66 | 16.3 | 20.9 | 5000 | 1000 |
| Re 0 | 12 | 631 | 3.91 | 28.8 | 72.9 | 1000 | 200 |
| Re 1 | 12 | 736 | 5.19 | 29.3 | 71.5 | 1000 | 200 |
| Re 2 | 11 | 725 | 3.82 | 29.2 | 73.0 | 1000 | 1000 |

### A.2 HYPERPARAMETERS

We use the following hyperparameters for all the datasets: epochs = 10, learning rate = 0.01, $n = 50$, $\epsilon = 0.003$, $v_0 = 0.6$, $v_1 = 0.3$, $v_2 = -0.05$, $v_3 = -0.1$, $v_4 = -0.3$, $v_{Tneg} = -1$, $\sigma = -1.2$, and $v_{Tpos} = 0.1$ or 0.35 or 0.8 depending on the dataset we investigate. For example, since there is no simple data in Re 2, we use $v_{Tpos} = 0.8$ instead to encourage R5 to memorize uncertain rules.

### A.3 IMPLEMENTATION DETAILS

We have mentioned the scoring scheme in Section 3.2. However, two optional hyperparameters are introduced, which can make sure the *ground truth* rules not be omitted due to large noise in datasets.

Recall that *simple* data are samples with a resolution length of 2, and the ground truth rules can naturally be collected from such data. To deduct such samples, R5 only needs one step. We give an extra *simple true reward* $v_{true}$ to the action to consolidate the corresponding rule in the memory. The other optional hyperparameter is a *simple wrong allowance* $v_{wrong}$. It can be applied when the rule collected from a simple sample is not in accordance with the existing rule in the rule memory.

Besides, to speed up training, we sort all the training samples by their maximum resolution paths, and train in the ascending order of the maximum resolution path in the first epoch. Thus, some ground truth rules can be quickly collected from simple data at an early stage.

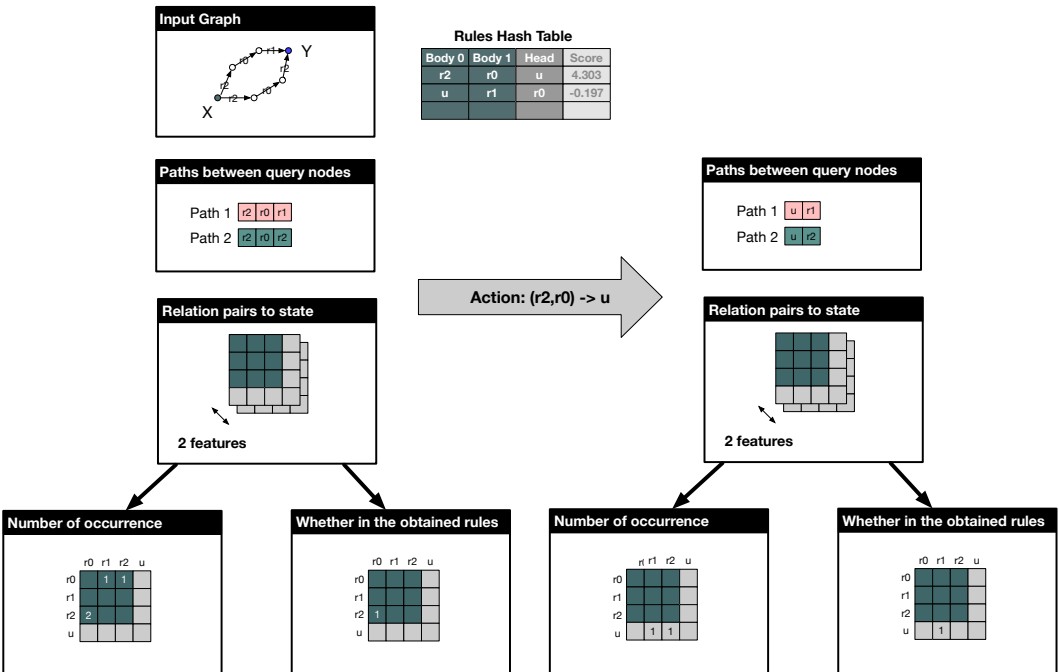

Figure 4: Illustration of the state and the state transition after taking an action.

## A.4 An Example of State Transition in the Decision-Making Process

Figure 4 gives a trivial example to illustrate how the state is constructed and the state transition after applying an action. As shown in Figure 2 and Section 2.1, the state represents $(m + n)(m + n)$ relation pairs and $k$ features. In this trivial example, we consider $m = 3, n = 1$ and $k = 2$ for a reasoning task, which means there are $m = 3$ known relations, and $n = 1$ allocated invented relation, thus in total $4 \times 4 = 16$ relation pairs. We also consider the two features to be *the number of occurrences* and *whether the relation pair is in the obtained rules*. Please be noted that we only compute the state features for the relation pairs in the paths. This is why $u, r_1 \rightarrow r_0$ is in the rules hash table but it is not counted when deciding whether a relation pair is in the obtained rules before taking the action.

## A.5 Extending to Complex Domains

**Handling $\hat{p} \leftarrow p_i$.** In Section 2 we explained that R5 realizes Horn clauses by firstly finding paths based on the query, and then omitting the entities and represent the long clauses by the composition of short clauses $u \leftarrow p_i \wedge p_j$. We want to make a note here R5 is also able to represent the very short clause $\hat{p} \leftarrow p_i$ by adding a dummy relation $r_{dm}$ to the state described in Section 2.1, and the corresponding predicates will be $p_{dm}$. Thus, $\hat{p} \leftarrow p_i$ can be represented by $\hat{p} \leftarrow p_i \wedge p_{dm}$. This has been implemented in the code.

**Triadic Relations and Hyper-Networks.** In this paper, we assume all rules are using the dyadic chain meta-rule template (Muggleton et al., 2015). However, R5 has the potential to handle triadic relations and hyper-networks. A hyper-network is formed by hyperedges, where a hyperedge refers to the relationship among data points that could go beyond pairwise, i.e., three or more objects are involved in each relation or predicate (Tu et al., 2018). Since R5 does not consider entities after paths are sampled, it naturally has the ability to handle hyperedges. For example, given two hyperedges $r_0(x_0, x_1, x_2)$ and $r_1(x_1, x_2, x_7)$, where $x_i$ are entities, part of the path can be $r_0, r_1$. However, hyperedges and hyper-networks is a different topic, which is not what we focus on in this paper. We have left this topic as future work.

## B Augmenting R5 with R-GCN

R5 makes predictions relying on the obtained rules. Different from the embedding-based methods, which can always return a relation type with the least loss, it is possible that R5 cannot give a known

relation as the prediction with its logical knowledge. During testing, R5 may fail to find the *body* of a deduction step in the memory, or finally deduce into an invented relation. Neither is able to give a valid prediction. We define the *invalid prediction ratio* as the ratio that R5 fails to return a known relation.

When a human is not able to make decisions based on logic, he may need the help of *intuition*. Similarly, we combine R5 with R-GCN, a GNN-based model, which is not able to explicitly reason to make predictions (similar to a human's intuition), to see if the results can be further improved when R5 is complemented by a GNN. Besides, R-GCN takes node embeddings into account, which may help to improve the predictions that require node properties.

Table 9: Results of augmenting R5 with R-GCN.

|             | Re 1      | Re 2      | World 32   |
|-------------|-----------|-----------|------------|
| R-GCN       | 0.250     | 0.371     | 0.596      |
| R5          | 0.590     | 0.440     | 0.899      |
| R5 + R-GCN  | **0.640** | **0.652** | **0.947**  |

The R-GCN model is trained with the following hyperparameters: a learning rate of 0.001, a node/relation embedding dimension of 200, 20 training epochs, 4 layers, and a dropout rate of 0.1.

Table 9 shows the results of augmenting R5 using R-GCN. And in the *R5+R-GCN* model, when R5 fails to give a valid prediction, we use R-GCN to predict the target relation. The invalid prediction ratios of R5 for Re 1, Re 2, and World 32 are 0.26, 0.434, and 0.085 respectively. As presented in the table, the prediction accuracy can further be improved by augmenting R5 with R-GCN, i.e., the accuracies improve by 0.05, 0.212, and 0.048 on the three datasets, respectively, indicating that the reasoning provided by R5 and GNN-based approaches are complementary to each other.

## C  RESULTS ON CLUTRR WITH HYPERPARAMETERS FINE-TUNED ON LONG STORIES

Table 10: Results on CLUTRR, trained on stories of lengths $\{2,3\}$ and evaluated on stories of lengths $\{4,...,10\}$. [*] means the numbers are taken from CTP's paper. [$_l$] means fine-tuned on long stories.

|            | 4Hops       | 5Hops       | 6Hops         | 7Hops         | 8Hops         | 9Hops         | 10Hops        |
|------------|-------------|-------------|---------------|---------------|---------------|---------------|---------------|
| R5         | .98±.02     | .99±.02     | .98±.03       | **.96±.05**   | **.97±.01**   | **.98±.03**   | **.97±.03**   |
| $CTP_L$*   | .98±.02     | .98±.03     | .97±.05       | .96±.04       | .94±.05       | .89±.07       | .89±.07       |
| $CTP_A$*   | .99±.02     | .99±.01     | **.99±.02**   | .96±.04       | .94±.05       | .89±.08       | .90±.07       |
| $CTP_M$*   | .97±.03     | .97±.03     | .96±.06       | .95±.06       | .93±.05       | .90±.06       | .89±.06       |
| GNTP*      | .49±.18     | .45±.21     | .38±.23       | .37±.21       | .32±.20       | .31±.19       | .31±.22       |
| $GAT_l^*$  | .92±.01     | .73±.04     | .56±.04       | .55±.04       | .54±.03       | .55±.04       | .50±.04       |
| $GCN_l^*$  | .84±.04     | .61±.03     | .51±.02       | .48±.02       | .45±.03       | .47±.05       | .41±.04       |
| $RNN_l^*$  | .93±.03     | .91±.03     | .79±.08       | .82±.06       | .75±.11       | .68±.07       | .64±.07       |
| $LSTM_l^*$ | 1.0±.00     | 1.0±.00     | .95±.03       | .94±.04       | .87±.08       | .86±.08       | .84±.09       |
| $GRU_l^*$  | .92±.05     | .88±.06     | .78±.09       | .77±.09       | .74±.08       | .66±.10       | .65±.08       |
| $CNNH_l^*$ | .94±.03     | .86±.06     | .77±.08       | .72±.08       | .64±.09       | .59±.10       | .59±.09       |
| $CNN_l^*$  | .93±.04     | .86±.07     | .84±.09       | .79±.08       | .77±.10       | .69±.09       | .70±.11       |
| $MHA_l^*$  | .81±.04     | .76±.04     | .74±.05       | .70±.04       | .69±.03       | .64±.05       | .67±.02       |

Table 11: Results on CLUTRR, trained on stories of lengths $\{2,3,4\}$ and evaluated on stories of lengths $\{5,...,10\}$. [*] means the numbers are taken from CTP's paper. [$_l$] means fine-tuned on long stories.

|            | 5 Hops      | 6 Hops      | 7 Hops        | 8 Hops        | 9 Hops        | 10 Hops       |
|------------|-------------|-------------|---------------|---------------|---------------|---------------|
| R5         | .99±.02     | .99±.04     | **.99±.03**   | 1.0±.02       | **.99±.02**   | **.98±.03**   |
| $CTP_L$*   | .99±.02     | .98±.04     | .97±.04       | .98±.03       | .97±.04       | .95±.04       |
| $CTP_A$*   | .99±.04     | .99±.03     | .97±.03       | .95±.06       | .93±.07       | .91±.05       |
| $CTP_M$*   | .98±.04     | .97±.06     | .95±.06       | .94±.08       | .93±.08       | .90±.09       |
| GNTP*      | .68±.28     | .63±.34     | .62±.31       | .59±.32       | .57±.34       | .52±.32       |
| $GAT_l^*$  | .98±.01     | .86±.04     | .79±.02       | .75±.03       | .73±.02       | .72±.03       |
| $GCN_l^*$  | .88±.01     | .78±.02     | .60±.02       | .57±.02       | .59±.04       | .51±.02       |
| $RNN_l^*$  | .96±.03     | .87±.09     | .82±.09       | .73±.09       | .65±.15       | .67±.16       |
| $LSTM_l^*$ | 1.0±.01     | .99±.02     | .96±.04       | .96±.04       | .94±.06       | .92±.07       |
| $GRU_l^*$  | .96±.02     | .88±.03     | .84±.04       | .79±.06       | .75±.08       | .78±.04       |
| $CNNH_l^*$ | 1.0±.00     | .99±.01     | .96±.02       | .91±.04       | .89±.04       | .87±.04       |
| $CNN_l^*$  | 1.0±.00     | .98±.01     | .97±.02       | .92±.03       | .89±.03       | .87±.04       |
| $MHA_l^*$  | .88±.03     | .83±.05     | .76±.04       | .72±.04       | .74±.05       | .70±.03       |

## D  CASE STUDIES

Table 12 and Table 13 present some examples of the recurrent relational reasoning process by R5 on CLUTRR and GraphLog datasets. For verification, GraphLog provides the ground truth rules sets used to generate each dataset, and in Tables 13 and 14 we mark such ground truth rules in bold. As presented, R5 is able to discover the ground truth rules, and learn long compositional rules. In the second instance in Table 13, we have an invented relation `unknown_44`. With this intermediate relation introduced, we can get a clause with three atoms in its body, i.e., `r_6←(r_9,r_3,r_9)`.

Table 14 shows some examples that R5 falsely predicts the queried relations. In the first example, both rules used are in the set of ground truth rules, leading to a prediction of `r_3` between the queried pair, which is different from the target relation `r_13` provided in the test set. Furthermore, we find that there is no ground truth rule in this world to allow the prediction of `r_13` by human deduction. Neither can R5 deduct `r_13`.

Table 12: Detailed breakdowns of deduction steps by R5 on CLUTRR.

| Target Relation | Deduction steps (head $\Leftarrow$ body) |
|---|---|
| Jason $\xrightarrow{sister}$ Evelyn | (Jason $\xrightarrow{daughter}$ Stephanie) $\Leftarrow$ (Jason $\xrightarrow{wife}$ Ruth, Ruth $\xrightarrow{daughter}$ Stephanie)
(Jason $\xrightarrow{sister}$ Evelyn) $\Leftarrow$ (Jason $\xrightarrow{daughter}$ Stephanie, Stephanie $\xrightarrow{aunt}$ Evelyn) |
| Jeff $\xrightarrow{aunt}$ Cristina | (Jeff $\xrightarrow{mother}$ Ruth) $\Leftarrow$ (Jeff $\xrightarrow{sister}$ Gloria, Gloria $\xrightarrow{mother}$ Ruth)
(Jason $\xrightarrow{sister}$ Cristina) $\Leftarrow$ (Jason $\xrightarrow{daughter}$ Stephanie, Stephanie $\xrightarrow{aunt}$ Cristina)
(Ruth $\xrightarrow{sister}$ Cristina) $\Leftarrow$ (Ruth $\xrightarrow{husband}$ Jason, Jason $\xrightarrow{sister}$ Cristina)
(Jeff $\xrightarrow{aunt}$ Cristina) $\Leftarrow$ (Jeff $\xrightarrow{mother}$ Ruth, Ruth $\xrightarrow{sister}$ Cristina) |
| Carmelita $\xrightarrow{son-in-law}$ Chuck | (Peter $\xrightarrow{sister}$ Brandi) $\Leftarrow$ (Peter $\xrightarrow{brother}$ Michael, Michael $\xrightarrow{sister}$ Brandi)
(Mark $\xrightarrow{sister}$ Brandi) $\Leftarrow$ (Mark $\xrightarrow{brother}$ Peter, Peter $\xrightarrow{sister}$ Brandi)
(Carmelita $\xrightarrow{grandson}$ Mark) $\Leftarrow$ (Carmelita $\xrightarrow{daughter}$ Martha, Martha $\xrightarrow{son}$ Mark)
(Carmelita $\xrightarrow{granddaughter}$ Brandi) $\Leftarrow$ (Carmelita $\xrightarrow{grandson}$ Mark, Mark $\xrightarrow{sister}$ Brandi)
(Carmelita $\xrightarrow{daughter}$ Elizabeth) $\Leftarrow$ (Carmelita $\xrightarrow{granddaughter}$ Brandi, Brandi $\xrightarrow{aunt}$ Elizabeth)
(Carmelita $\xrightarrow{son-in-law}$ Chuck) $\Leftarrow$ (Carmelita $\xrightarrow{daughter}$ Elizabeth, Elizabeth $\xrightarrow{husband}$ Chuck) |

Table 13: Detailed breakdowns of deduction steps by R5 on GraphLog.

| Target Relation | Deduction steps |
|---|---|
| r_16 | **r_2 $\Leftarrow$ (r_3, r_13)**
**r_1 $\Leftarrow$ (r_2, r_20)**
**r_9 $\Leftarrow$ (r_3, r_6)**
**r_5 $\Leftarrow$ (r_1, r_9)**
**r_7 $\Leftarrow$ (r_5, r_6)**
r_16 $\Leftarrow$ (r_2, r_7) |
| r_6 | **r_2 $\Leftarrow$ (r_3, r_13)**
**r_1 $\Leftarrow$ (r_2, r_20)**
r_3 $\Leftarrow$ (r_1, r_20)
unknown_44 $\Leftarrow$ (r_3, r_9)
r_6 $\Leftarrow$ (r_3, r_3)
**r_9 $\Leftarrow$ (r_3, r_6)**
r_6 $\Leftarrow$ (r_9, unknown_44) |

Table 14: Wrong relation predictions by R5 on GraphLog.

| Target Relation | Prediction | Deduction steps |
|---|---|---|
| r_13 | r_3 | **r_6 $\Leftarrow$ (r_20, r_13)**
**r_3 $\Leftarrow$ (r_16, r_6)** |
| r_5 | r_18 | **r_19 $\Leftarrow$ (r_15, r_8)**
**r_17 $\Leftarrow$ (r_19, r_4)**
**r_18 $\Leftarrow$ (r_8, r_17)** |

In the second example, all the three rules learned by R5 are ground truth rules, enabling us to predict r_18 from the resolution path [r_8, r_15, r_8, r_4]. Yet, r_18, although correct, does not exactly match the target relation r_5 provided by the test set. We note that there is not any ground truth rule provided by the dataset whose head is r_5.

However, R5 has also learnt an additional rule from the training samples headed to r_5:

$$r\_5 \leftarrow (r\_8, r\_18), \tag{5}$$

which is not in the set of ground truth rules. With Rule (5), we have another sequence of deductions that can lead to r_5 from the resolution path [r_8, r_15, r_8, r_4]:

$$\begin{aligned}
\mathbf{r\_10} &\leftarrow \mathbf{(r\_8, r\_4)}, \\
\mathbf{r\_18} &\leftarrow \mathbf{(r\_15, r\_10)}, \\
r\_5 &\leftarrow (r\_8, r\_18),
\end{aligned} \tag{6}$$

where the first two rules in bold are in the ground truth rules, while the third is not.

Since there are two valid relation types `r_5` and `r_18` between the queried node pair, R5 may not be able to tell which one should be selected without knowledge of the entities. This is an example where the entity properties may help with logical inference.

## E    DISCUSSIONS ON SOME KB REASONING WORKS

As discussed in Section 4, KB reasoning is not what we focus on. However, R5 has some strength compared with some of the existing KB relational learning works. Yang et al. (2017) learns the rules in the form `query(Y,X)` $\leftarrow$ `R_n(Y,Z_n)` $\land \cdots \land$ `R_1(Z_1,X)`, where each predicate strictly contains 2 entities. Whereas, R5 is able to consider predicates with any number of entities as explained in Appendix A.5.

Moreover, many RL Path-finding Agents (Xiong et al., 2017; Chen et al., 2018; Das et al., 2017; Lin et al., 2018; Shen et al., 2018) are proposed for KB reasoning. The major challenges for these works include the sparse rewards. For example, at each step of decision-making, Shen et al. (2018) selects neighbouring edges at the current node, and follows the edges to the next node. Termination occurs when the agent gives a "STOP" action instead of selecting the next neighbouring edge. The termination criterion is unclear, which is one of the important reasons for the sparse reward. In contrast, the MCTS in R5 guarantees that only the feasible relation pairs can be selected. Termination occurs when reduced to a single relation. The criterion is clear.

However, R5 faces some challenges to generalize to KB reasoning. Apart from the ones mentioned in Section 4, currently, R5 does not generate probabilities or scores for a prediction, which is not capable of KB Completion since KB Completion is actually a set of ranking tasks. Besides, R5 requires pre-sampling for the paths that entails the query. Since all the triples in a Knowledge Graph share the same *training graph*, which is usually very large, it will take a long time to enumerate all the paths. If we choose not to perform enumeration and sample some paths instead, the sampling method is also a topic that needs to be investigated. Due to these challenges, we leave the application to KB as future work.

## F    COMPARISON WITH NEURAL-LP AND RNNLOGIC

Neural-LP (Yang et al., 2017) is a neural logic programming method and RNNLogic (Qu et al., 2021) is a principled probabilistic method, where both of the two approaches are designed for Knowledge Base Completion (KBC) problems. These methods requires enumeration of all possible rules given a max rule length $T$. Thus, the complexity of these models grows exponentially as max rule length increases, which is an outstanding disadvantage for systematicity problems.

Table 15: Comparison with Neural-LP and RNNLogic.

| Model | GraphLog World 17 | GraphLog Re 0 |
|---|---|---|
| R5 | 0.947 | 0.665 |
| Neural-LP | 0.057 | 0.080 |
| RNNLogic w/o emb | 0.606 | 0.348 |

We tweaked the code of RNNLogic, which was originally designed for entity ranking, to predict and rank relation r given $(h, ?, t)$ on GraphLog. The results are shown in Table 15. One issue we observed was that we were only able to run RNNLogic with max rule length $T$ up to 7 on a powerful server, beyond which there was out-of-memory issue. (For instance, setting $T = 6$ for RNNLogic on GraphLog Re 0 requires more than 130 G memory.) We report its best result in Table 15, which is achieved when $T = 5$. It is shown that R5 substantially outperforms RNNLogic, which is in fact not designed for this task. Similarly we revised NeurlLP to make it work for the relation prediction prediction task. But unfortunately, it didn't work well with a relation prediction accuracy of less than 10%.

## G EXTRA FEATURES TO RECORD THE MAX/MIN NUMBER OF OCCURRENCES AMONG PATHS.

Table 16: Comparison with max/min number of occurrences among path added to features.

| Model | GraphLog World 17 | GraphLog Re 0 |
|---|---|---|
| R5 | 0.947 | 0.665 |
| R5 with extra features | 0.941 | 0.673 |

## H NUMBER OF PRE-ALLOCATED INVENTED RELATIONS

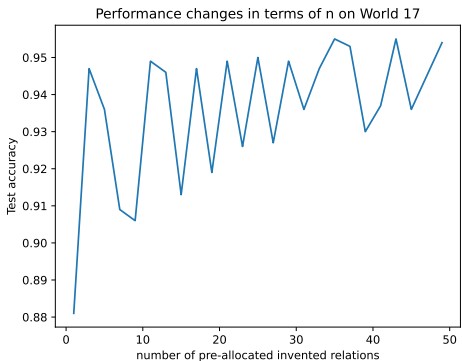 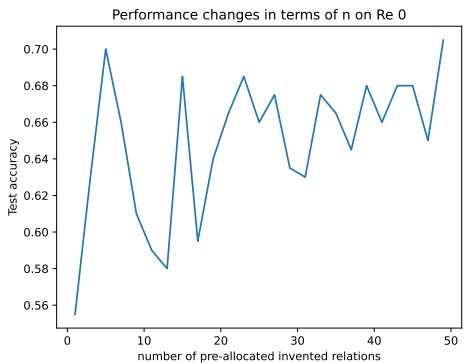

Figure 5: The test accuracy as the number of pre-allocated invented relations varies for World 17 in GraphLog.

Figure 6: The test accuracy as the number of pre-allocated invented relations varies for Re 0 in GraphLog.

In this paper, we have introduced $n$ pre-allocated invented relations to capture the intermediate short rules during the decision-making procedure. For example, $r_1, r_2, r_3 \to r$ can be decomposed into $(r_1, r_2 \to \text{invented}_0)$, $(\text{invented}_0, r_3 \to r)$. The details can be found in Section 2.2. To choose $n$, we conduct experiments on two GraphLog datasets, World 17 and Re 0, as $n$ varies. We use the parameters listed in Appendix A.2, except that 5 training epochs are used instead of 10 to speed up training. As presented in Figure 5 and Figure 6, R5 is able to get a near optimal performance even when $n$ is small. But as $n$ becomes larger, there is a general trend that the performance can be more stable. Intuitively, a small $n$ makes the dynamic memory replace the invented (unknown) relations more frequently as new invented relations appear, slowing down the convergence (as we discussed in Section 3.3). Therefore, in our experiments, as computational resources permit, we set $n$ to be 50 for all the datasets. However, if a small $n$ is required in practice, we can also use a validation set to select some $n$ that is sufficient for each specific dataset.

