# OpenReview forum: "R5: Rule Discovery with Reinforced and Recurrent Relational Reasoning"
_ICLR.cc/2022/Conference — ICLR 2022 Spotlight_

### Official Review · Reviewer_dUTD · 2021-10-31

**Correctness:** 4
**Technical Novelty And Significance:** 3
**Empirical Novelty And Significance:** 3
**Recommendation:** 6
**Confidence:** 5

**Main Review:**

Program induction is a hard problem, especially for large-scale tasks in noisy
domains. This problem can be formulated as a Relational Pathfinding task [1], which
has been studied for many years in the area of Inductive Logic Programming. In
Statistical Relational Learning there is also a missing related work that
searches for grounded relational paths first and then applies machine learning
to extract first-order logical rules [2].

Compared to previous works in this direction, the main contribution of this
work is applying reinforcement learning in the relational pathfinding problem. The
proposed approach works surprisingly well in the experiments, and the combination of
neural-guided heuristic learning for MCTS and the symbolic rewarding method perform good
in the relational pathfinding problem.

The main weakness of relational pathfinding is the grounding problem, i.e., for
each training example, the algorithm needs to search for ground paths before
training. Besides, the pathfinding algorithm may suffer from incompleteness if
the hypothesis has some complex structures, e.g., if there are some useful
relations do not showing in a path (node "z" in p(x,y)<-q(x,y),r(x,z)), or if
there exists higher-order primitive predicates or triadic predicates. Another
possible weakness is that reinforcement learning usually requires large numbers
of training epochs, especially when learning the heuristics for program
induction.

### Some minor issues:
- "unknown relations" should be called "invented predicates/relations", following the
  convention in the area of program induction;
- The algorithm implicitly assumes all rules are using the same template, which
  is a dyadic chain meta-rule [3], which could result in incompleteness [4]
  (i.e., some hypotheses can never be learned), I think this should be made
  clear in this paper.
- Following my last point, the proposed method seems can only learn dyadic chain
  rules in datalog (e.g., every rule in the target hypothesis have the form of
  "p(x,y)<-q(x,z), r(z,y)", and the language is function-free). Is it possible to extend it to complex domains?
- The learned policy can not be transferred to other tasks (e.g., adapt from one
  task to another task, or from one target relation to another one).

### References:
1. Bradley L. Richards and Raymond J. Mooney. Learning relations by pathfinding.
   In Proceedings of the 10th National Conference on Artificial Intelligence,
   pages 50–55, San Jose, CA, July 1992.
2. Wang-Zhou Dai and Zhi-Hua Zhou. Statistical unfolded logic learning. In:
   Proceedings of the 7th Asian Conference on Machine Learning (ACML’15), Hong
   Kong, 2015, JMLR: W&CP 45, pp. 349-361.
3. S.H. Muggleton, D. Lin, N. Pahlavi, and A. Tamaddoni-Nezhad.
   Meta-interpretive learning: application to grammatical inference. Machine
   Learning, 94:25-49, 2014.
4. Andrew Cropper and Stephen H. Muggleton. Logical minimisation of meta-rules
   within meta-interpretive learning. In Proceedings of the 24th International
   Conference on Inductive Logic Programming, pages 65-78.
   Springer-Verlag, 2015. LNAI 9046.


**Summary Of The Paper:**

This paper presents a novel method for rule induction. The main idea is to apply
reinforcement learning in the task of relational pathfinding within a finite
Herbrand base. The reinforcement learner uses MCTS to find the best routes to
establish the path in between the two arguments of training examples, while the
useful actions (which are length-two relational paths) are maintained in a hash
table as the induced ruleset. Experimental results has shown the effectiveness
of the proposed method.

**Summary Of The Review:**

This paper presents a novel algorithm to solve program induction formulated as a
relational pathfinding problem. The proposed approach outperforms existing rule
induction methods in several benchmark datasets. It has missed several important
references and needs further clarification, so I would like to recommend to
accept if the authors can fix those problems.

---

> ### Author Response · Authors · 2021-11-18
> **Response to Reviewer dUTD**
>
> Your suggestions are very helpful to us. Thank you very much! According to your suggestions, we made the following modifications to our manuscript:
>
> - We replaced the *unknown relations* with *invented relations* in our paper.
> - We cited the suggested references on program induction and relational pathfinding. We revised the Related Work section in the paper and clarified (in red).
> - We updated the explanation in A.5 for higher-order relations.
>
> To address your questions, we present the following responses:
>
> **Q1** Is it possible to extend it to complex domains?
>
> Yes, it is. As we discussed in Appendix A.5, since our framework does not consider entities at all, it is free to have whatever number of entities in a predicate. Thus, our framework has the potential to extend to hyper-networks, where higher-order primitive predicates or triadic predicates may exist. For example, given a query *(x0,r=?,x7)*, we can reason about *r* with the rule: *r(x0, x7) <- r0(x0,x1,x2), r1(x1,x2,x7)*, where *x_i* are entities and *r0(x0,x1,x2), r1(x1,x2,x7)* are two hyper-edges.
>
> Besides, it is also possible to handle cases like *“node ‘z’ in p(x,y)<-q(x,y),r(x,z)”*, by adding a reverse relation for each relation, like what recent Knowledge Base models usually do (For example, Neural-LP and RNNLogic). For example, given *q(x,y)*, *r(x,z)*, we can also get *q^{-1}(y,x)* and *r^{-1}(z,x)*, then a path can be found: *p(x,y) <- r(x,z), r^{-1}(z,x), q(x,y)*.
>
> **Q2** Reinforcement learning usually requires large numbers of training epochs, especially when learning the heuristics for program induction.
>
> Yes, we agree that RL usually requires large numbers of training epochs. However, with the help of our proposed dynamic memory module and the associated state features, R5 actually converges very quickly as presented in the experiments (Refer to Figure 3). We also mentioned in Appendix A.2 that all the results of R5 in this paper were obtained from experiments run by only 10 epochs.
>
> **Q3** Can the learned policy be transferred to other tasks (e.g., adapt from one task to another task, or from one target relation to another one)?
>
> If we understand your question correctly, we do not consider transfer learning in this paper. However, it is presented in our paper (refer to Section 3) that after training on one task, our framework can generalize to a different task with longer relational path and different entities but a similar rule space.
>
> We hope our clarifications answer your questions. Thank you for your comments.

---

> ### Author Response · Authors · 2021-11-18
> **New revision to the paper and response**
>
> We added some clarifications in Section 4 (related work) and Appendix A.5 of the paper (Still in red). We also edited the response.

---

### Official Review · Reviewer_Mm5p · 2021-11-01

**Correctness:** 3
**Technical Novelty And Significance:** 3
**Empirical Novelty And Significance:** 3
**Recommendation:** 6
**Confidence:** 3

**Main Review:**

In general the proposed approach is reasonable and effective for the targeted 2 tasks. The introduction of new relations, and application of RL are interesting. However, the task and model design seem to be very specific to a type of task where the relevant part of a graph (to the prediction) is restricted to a path, which can be recursively derived from the target relation. The separation of path finding and reasoning also seems to be an artifact from the specific task assumption.


Clarity related questions:
It would be helpful to formally define the relation prediction task.

It would be helpful to describe the path sampling procedure.

How is the model updated given the loss function l?

The state representation has one entry per relation pair. Will that lead to information loss when a relation pair appears multiple times in the path?

"recurrent reasoning agent enables us to predict ... efficiently" -- it is more efficient than what approach? and why?

Table 4 and 5: given that the rule recalls of R5 are almost 100% why the accuracy much lower than 100%?



**Summary Of The Paper:**

This work explores a rule learning approach (R5) for 2 relation prediction tasks (CLUTRR and GraphLog). The proposed approach starts by finding connecting paths between the two query entities. Then it recursively merge relation pairs until it consists of a single relation output. The merging process is controlled by a policy/value network trained with episodic rewards. The rules are induced whenever the output relation matches the gold relation -- in a fashion similar to that of curriculum learning.

Experiments show that R5 generalizes better than previous approaches such as Graph Attention Networks (GAN) and Conditional Theorem Provers (CTPs) especially when generalizing to paths longer than those from the training set.


**Summary Of The Review:**

An interesting approach with good results for a limited task setup.

---

> ### Author Response · Authors · 2021-11-18
> **Response to Reviewer Mm5p**
>
> Thank you very much for your valuable review! We made the following modifications to our paper according to your suggestions:
>
> - We formally define the relation prediction task in Section 2. (in green)
> - As we described in Section 2.1, our path sampling procedure is as follows: 1) for the datasets that have very few paths for each query, we perform path enumeration; 2) otherwise, we **randomly** sample L paths. Our path sampling procedure is simple but effective as demonstrated by the experiments. In our future work, we plan to develop a more sophisticated sampling procedure to handle more complex situations. For example, we can learn a path finder by graph neural networks and merge it with the reasoner, and learn the two modules in an end-to-end manner.
>
> For your questions or concerns:
>
> **C1** The task and model design seem to be very specific to a type of task where the relevant part of a graph (to the prediction) is restricted to a path.
>
> In fact, our relational path-finding and reasoning task is actually program induction, which is a hard problem that has been studied for many years in the area of Inductive Logic Programming and Statistical Relational Learning, especially for large-scale tasks in noisy domains (thanks to Reviewer dUTD for pointing this out). We added clarification about this in Section 2 in our paper (in green).
>
> Besides, when the reasoning process relies on a sub-graph instead of a simple path, we can actually transform it into a path by adding a reverse relation for each relation (as we described in detail in our response to Reviewer dUTD). Moreover, R5 has the potential to handle more complex situations (e.g., there exists higher-order primitive predicates or triadic predicates) by introducing hypergraph representations, where a connection between two or more vertices of a hypergraph is denoted by a hyper-edge. In this way, we can sample hyper-paths and reason over them to handle more complex situations.
>
> **Q1** How is the model updated given the loss function l?
>
> The loss function is the sum of: 1) the mean-squared error of the value; and 2) the cross-entropy loss of the policy. The parameters theta of the neural network are adjusted by gradient descent on this loss function l.
>
> **Q2** The state representation has one entry per relation pair. Will that lead to information loss when a relation pair appears multiple times in the path?
>
> As we mentioned in Section 2.1, the number of occurrences of a relation pair is one of the state features, and the model keeps track of it. If a relation pair appears multiple times in the path, it will be counted to this feature. Thus, this information is not lost since we keep tracking it. It is also possible to add two extra features to record the max/min number of occurrences among paths. We conduct this experiment on a couple of datasets of GraphLog, and the result is added to Appendix G. As is shown, these two features do not influence the performance much. Therefore, we did not include it in our previous experiments.
>
> **Q3** "recurrent reasoning agent enables us to predict ... efficiently" -- it is more efficient than what approach? and why?
>
> It is more efficient than only using the MCTS. In Section 2.1, we introduced how the neural networks guide the search of the Monte Carlo Tree. A sequential decision-making task can be performed with only the MCTS. The neural networks can further store the Q(value at the nodes in the tree) and P(policy/probability distribution of child node selection) in the previously traversed trees. Thus, Q and P can be initialized accordingly at the following MCTS, and perform sequential decision-making tasks efficiently.
>
> **Q4** Table 4 and 5: given that the rule recalls of R5 are almost 100%, why is the accuracy much lower than 100%?
>
> The reason is that Table 4 and Table 5 are the results on **noisy** dataset GraphLog as we clarified in Section 3.2. For the results on the clean data, referring to Section 3.1, R5 gets nearly 100% accuracy on CLUTRR.
>
> We hope our clarifications address your questions. Again, thank you for your comments. We have added the above clarifications in green to the paper.

---

### Official Review · Reviewer_TLJz · 2021-11-04

**Correctness:** 3
**Technical Novelty And Significance:** 3
**Empirical Novelty And Significance:** 3
**Recommendation:** 5
**Confidence:** 3

**Main Review:**


Strengths

S1 The dynamic rule memory module is interesting.

S2 Detailed ablation study and case study.

S2 The paper is well written and well organized.

Weaknesses

W1 RL agent aims to seek long-term and maximum overall reward to achieve an optimal solution. However, this paper aims to divide the long-term decision into independent short-term decision. Therefore, I don’t see why RL should be used in this scenario.

W2 Too much reliance on heuristics. For example, it utilizes statistical features, such as the number of occurrences to represent relation pairs.

W3 How to recover the logical rule is unclear to me. Should we enumerate over the whole rule space to select the rules with the highest score?

W4 How do the authors handle uncertainty during the deduction is unclear. It is possible for a two relation sequence to have multi heads. For example, given two relation sequences hasAunt and hasSister, both hasMother and HasAunt can be the head. How can the proposed solution solve the multi head scenario?

W5 The author should discuss more about how to choose the value of n (number of unknown relations). Intuitively, a small n may reduce the expressiveness of the model while a big n may cause computational inefficiency. Choosing the value of n could be a tricky problem.

W6 Lack of some representative baselines. For example, NeuralLP, RNNLogic are missing.

W7 Datasets include too few relations.To show the scalability of the proposed model, it will be better to conduct experiments on datasets with more relations e.g., FB15k.

**Summary Of The Paper:**

The author investigates the rule learning problem and  proposes R5, a reinforcement learning  based framework to reason over relational graph data and explicitly mines underlying compositional logical rules from observations. It mainly contains two components: RL agent to model rule learning as a sequential decision-making problem and a dynamic rule memory module to maintain and score candidate rules during the training of the reasoning agent.


**Summary Of The Review:**

The author investigates an interesting problem, logical rule learning. It also proposes an interesting idea to decompose the long path into small pieces to recurrently learn rules. Although the idea is interesting, the model and the experiments are less convincing as pointed out in the weaknesses. It will be better for the author to clearly illustrate the reasoning why they choose the  RL model to learn rules and how they handle uncertainty during the deduction. They also need to include more baselines and larger datasets to make the experiments more convincing.

---

> ### Author Response · Authors · 2021-11-18
> **Response to Reviewer TLJz**
>
> Thank you for the valuable suggestions!
> For your questions:
>
> **W1)** The RL module in our framework is similar to playing a board game, e.g., Go, using an RL agent. [1] In a Go game, the RL agent learns a policy network that seeks to maximize long-term overall reward (i.e. winning the game) by making one move at each step. Similarly, in this paper, we aim to divide the long-term objective (giving a correct prediction) into short-term decisions to optimize the overall reward. In AlphaGo Zero, the policy network is trained by a self-play reinforcement learning algorithm. Here, in R5, we train an RL agent for sequential path-finding combined with a dynamic rule discovery and recording procedure to give correct predictions between nodes, despite the long path between them.
>
> **W2)** We only use 8 simple statistical features to characterize each pair of relations, e.g., the number of occurrences, the index of the most viewed position among the paths, whether the relation pair is in the obtained rules, whether the relation pair consists of two known relations, whether the rule head is a known relation if the pair is in the rule memory, and the score of the corresponding rule if the pair is in the rule memory, etc. These features characterize the graph topological information and are easy to obtain, with very low cost. And they are effective, since R5 substantially beats prior art (e.g., GNN based and other reasoning-based schemes) in relation prediction tasks. One can certainly use, e.g., pretrained GNN to extract more complex features to be added to state modeling, which also incurs higher cost and complexity.
>
> **W3)** R5 does not enumerate over the whole rule space. The term “rule space” usually refers to the set of all possible rules. Note that the rules in our paper can be very long, e.g., r1, r2, r3, …, rn-> r. We do not enumerate all such long rules and learn their scores to make inference. Instead, R5 considers only short rules in the form of *(p,q)->r*, and a long compositional rule is derived as a sequence of such short rules during sequential decision-making. For example, r1, r2, r3-> r can be decomposed into (r1, r2->unknown1), (unknown1, r3->r).  Given a graph, at each step of the decision-making procedure, R5 firstly needs to decide a relation pair (p,q) in the graph to combine using MCTS and the policy value network. And then R5 will lookup (p,q) in the dynamic memory as shown in Figure 2 to find the head relation to combine the pair (p,q) into. If (p, q) does not exist in the rule memory,  R5 will introduce an unknown relation to create a new rule (p,q)->unknown1. Unknown relations can also be recovered and replaced by existing relations during the backtrack rewriting mechanism (Algorithm 1) in our Dynamic Rule Memory, which is one of our main novelties (see Sec. 2.2.). Unimportant rules may also be deleted. This is in contrast to some prior rule extraction method (e.g., NeurlLP), which has to enumerate exponentially many rules and learn their scoring, which will fail as the rule length grows to more than 2 or 3. The length of rules we consider could be up to 10 in CLUTRR and 15 in GraphLog.
>
> **W4)** Thank you for pointing it out. We do not consider multi-head scenario in this paper.  The relational prediction tasks on the datasets in this paper have only a single target relation. Therefore there are no multi-head cases. However, we have added a formal definition of *relation prediction* task in Section 2 to clarify the problem solved in this paper (In green). In future, a natural extension could be the multi-label problem, for which we can allow multiple rules (each associated with a score) for each relation pair in the dynamic memory, which serves as a dictionary.
>
> **W5)** Thank you for pointing it out. We conducted extra experiments and added two figures in Appendix H to illustrate the performance as number of unknown relations *n* varies on two GraphLog datasets. These experiments are conducted with the parameters in Appendix A.2. As presented in the figures, R5 is able to get a near optimal performance even when *n* is small. But as *n* becomes larger, there is a general trend that the performance can be more stable. Intuitively, a small *n* makes the dynamic memory replace the invented (unknown) relations more frequently as new invented relations appear, slowing down the convergence (as we discussed in Section 3.3). Therefore, in our experiments, as computational resources permit, we set *n* to be 50 for all the datasets. However, if a small *n* is required in practice, we can also use a validation set to select some *n* that is sufficient for each specific dataset.

---

> > ### Author Response · Authors · 2021-11-18
> > **Response to Reviewer TLJz Cont. (1)**
> >
> > **W6)** NeuralLP and RNNLogic are models designed for transductivity, whereas we focus on the systematicity in our paper. Recall that strong systematicity means the model is able to train on short-rule samples and generalize to long-rule samples (like the example in Figure 1). In our systematicity-testing tasks, the length of the compositional rules used to perform prediction can be up to 15 (Table 2,3,4 & 5), which is a challenging case for the current literature to solve. For example, since NeuralLP requires enumeration of all possible rules according to Eq. 5-8 in their paper, NeuralLP needs to consider 20^16=6.5536e20 rules (i.e. **the complexity of these models grows exponentially as max rule length increases**). Please also note that in the NeuralLP paper, the max length is set to be 2 or 3 for experiments on all the datasets. Besides, it has been presented and shown in the CTP’s paper that CTP is better than NeuralLP. We have taken the better model CTP as our baseline, and shown that R5 beats CTP. Therefore, models like NeuralLP were not treated as our baselines in our paper. Similarly, RNNLogic also suffers from the same out-of-memory issue as rule length grows, when performing rule discovery and generation.
> >
> > However, during the rebuttal opportunity, we have conducted extra experiments for NeuralLP and RNNLogic on two GraphLog datasets. We tweaked the code of RNNLogic, which was originally designed for entity ranking, to predict r given (h,?,t) on GraphLog. The results have been added to Appendix F. One issue we observed was that we were only able to run RNNLogic with max rule length T up to 7 on a powerful server, beyond which there was out-of-memory issue. (Running RNNLogic on GraphLog Re 0 dataset with T = 6 takes around 130G memory.) We report its best result in the revised manuscript, which is achieved when T=5. It is shown that R5 substantially outperforms RNNLogic. Similarly we revised NeurlLP to make it work for the relation prediction task. But unfortunately, it didn’t work well with a relation prediction accuracy of less than 10%.
> > In contrast to these methods, by using a policy value network learned by RL and Monte Carlo Tree Search to decide each step of reasoning, R5 achieves superior systematicity and can generalize to much longer rules in a larger graph, even if trained on small graphs with short rules.

---

> > > ### Author Response · Authors · 2021-11-18
> > > **Response to Reviewer TLJz Cont. (2)**
> > >
> > > **W7)** First, FB15k is a knowledge base completion (KBC) dataset. The author of RNNLogic has explained the term “Link Prediction” in KBC literature in Section 4 in their paper as well as their response to AnonReviewer4 during the author rebuttal period of ICLR 2021, which can be found on OpenReview:
> > >
> > > > *“In a link prediction task, the rank of entities is computed. To be more specific, given a query (h, r, ?), the task aims to infer the correct entity t, so that entity h and entity t have the relation r.”*
> > >
> > > In other words, this is an entity ranking problem: given *h* and *r*, what is the *t* they are linking to? This task is different from the Relation Prediction task considered in our paper, where the query is a pair of nodes *(X,Y)*, and the task is to predict the missing relation *r* between *X* and *Y*. Therefore, as we claimed in Section 4 and Appendix E, we do not consider KBC problems in this paper.
> > >
> > > Second, as we claimed in our paper, our framework focuses on the systematicity instead of the transductivity. Systematicity [3] requires the model to train on data with short rules and be able to generalize to problems with long rules. In contrast, transductivity refers to generalizing to different entities. The two problems are different. If we learn how to infer the son of someone, we also know how to infer the son of someone else. This is the entity ranking (link prediction) problem (h,r,t=?) and needs the ability of transductivity. Traditional methods designed for KBC can handle this task, because short rules are usually sufficient to find the answer. But they fail to generalize in a systematic way or demonstrate robust reasoning ability, which stems from the classical AI challenge of Inductive Logic programming (ILP) dated back to 1990. We are dealing with systematicity, i.e., what is the relationship between two (probably far apart, e.g., 15 hops away) nodes in a complex graph, and can we generalize and adapt to long rules required to discover such a relationship?
> > >
> > > Specifically, CLUTRR is a semi-synthetic benchmark designed to explicitly test a Natural Language Understanding (NLU) model’s ability for systematic and robust logical generalization. Given a story in a paragraph of text, the query is to infer the relationship between two characters in the story.  The key challenge is to find the potentially long logical path between characters. GraphLog is a benchmark designed to test the effectiveness of relational learning algorithms, such as GNNs and inductive methods to adapt to new tasks. These are new datasets serving a different purpose than traditional knowledge base. When long compositional rules are present, existing GNNs and rule mining methods fall short on these challenging tasks. We aim to fill in the gap to focus on inductive logical reasoning and the ability of systematic generalization.
> > >
> > > Lastly, we agree with the reviewer that R5 may also have potential to be extended to KB entity ranking problems in the future. We hope the framework and insights presented in this paper could help the whole community in the interesting research direction to develop true logical reasoning and generalization capabilities.
> > >
> > > [1] David Silver, Thomas Hubert, Julian Schrittwieser, Ioannis Antonoglou, Matthew Lai, Arthur Guez, Marc Lanctot, Laurent Sifre, Dharshan Kumaran, Thore Graepel, Timothy Lillicrap, Karen Simonyan, and Demis Hassabis. Mastering chess and shogi by self-play with a general reinforcement learning algorithm, 2017
> > >
> > > [2] Michael Schlichtkrull, Thomas N Kipf, Peter Bloem, Rianne Van Den Berg, Ivan Titov, and Max Welling. Modeling relational data with graph convolutional networks. In European semantic web conference, pp. 593–607. Springer, 2018.
> > >
> > > [3] Dieuwke Hupkes, Verna Dankers, Mathijs Mul, and Elia Bruni. Compositionality decomposed: how do neural networks generalise?Journal of Artificial Intelligence Research, 67:757–795, 2020.

---

### Decision · Program_Chairs · 2022-01-20

**Decision:**

Accept (Spotlight)

**Comment:**

A novel method is described that uses RL to search for a rule set which predicts multiple relations at once for KBC-like problems.  The rules can include latent predicates, which reduces the complexity of individual rules, similar to Cropper & Muggleton's (2015) meta-interpretive learning framework, which is usual for rule-learning systems.  Another novel aspect is use of a cache memory for rules.

Pros
 - the idea of using RL instead of carefully-designed discrete search for symbolic learning systems is a very nice novel idea
 - the experimental results are strong

Cons
 - the benchmarks are synthetic (although GraphLog does at least include noise)
 - although the Cropper and Muggleton work is cited the relationship between the search spaces of the two systems is not discussed - this should be corrected in the final version.